# On Federated Compositional Optimization: Algorithms, Analysis, and Guarantees

**Prashant Khanduri[†], Chengyin Li[‡], Rafi-Ibn Sultan[†], Aditi Sarker[†],**
**Yao Qiang[◇], Joerg Kliewer[*], Dongxiao Zhu[†]**

[†]**Wayne State University,** [‡]**Henry Ford Health,** [◇]**Oakland University,**
[*]**New Jersey Institute of Technology**

*khanduri.prashant@wayne.edu, cli6@hfhs.org, rafis@wayne.edu, hq1351@wayne.edu, qiang@oakland.edu,*
*joerg.kliewer@njit.edu, dzhu@wayne.edu*

**Reviewed on OpenReview:** *https://openreview.net/pdf?id=4uRlbSNevR*

## Abstract

Compositional optimization (CO) has recently gained popularity due to its applications in many machine learning applications. The large-scale and distributed nature of data necessitates efficient federated learning (FL) algorithms for CO, but the compositional structure of the objective poses significant challenges. Current methods either rely on large batch gradients (which are impractical), require expensive computations, or suffer from suboptimal guarantees. To address these challenges, we propose efficient FedAvg-type algorithms for solving non-convex CO in the FL setting. We first theoretically establish that standard FedAvg fails in solving the federated CO problems due to data heterogeneity, which amplifies bias in local gradient estimates. Our analysis shows that controlling this bias necessarily requires either *additional communication* or *additional structural assumptions*. To this end, we develop two algorithms for solving the federated CO problem. First, we propose FedDRO that utilizes the compositional problem structure to design a communication strategy that allows FedAvg to converge. FedDRO achieves a sample complexity of $\mathcal{O}(\epsilon^{-2})$ and a communication complexity of $\mathcal{O}(\epsilon^{-3/2})$ when the inner compositional objective is low-dimensional. When the inner objective is high-dimensional, the communication complexity increases to $\mathcal{O}(\epsilon^{-2})$, while the sample complexity remains $\mathcal{O}(\epsilon^{-2})$. Then we propose DS-FedDRO, a two-sided learning rate algorithm that leverages an additional assumption to improve upon the communication complexity of FedDRO. DS-FedDRO achieves the optimal $\mathcal{O}(\epsilon^{-2})$ sample and $\mathcal{O}(\epsilon^{-1})$ communication complexity irrespective of the dimensionality of the inner compositional objective. We corroborate our theoretical findings with empirical studies on large-scale CO problems.

## 1 Introduction

Compositional optimization (CO) problems deal with the minimization of the composition of functions. A standard CO problem takes the form

$$\min_{x \in \mathbb{R}^d} f(g(x)) \quad \text{where} \quad g(x) := \mathbb{E}_{\zeta \sim \mathcal{D}_g}[g(x; \zeta)], \tag{1}$$

where $x \in \mathbb{R}^d$ is the optimization variable, $f : \mathbb{R}^{d_g} \to R$ and $g : \mathbb{R}^d \to \mathbb{R}^{d_g}$ are smooth functions, and $\zeta \sim \mathcal{D}_g$ represents a stochastic sample of $g(\cdot)$ from distribution $\mathcal{D}_g$. CO finds applications in a broad range of machine learning applications, including but not limited to distributionally robust optimization (DRO) Qi et al. (2022), meta-learning Finn et al. (2017), hyperparameter tuning Franceschi et al. (2018), phase retrieval Duchi & Ruan (2019), portfolio optimization Shapiro et al. (2021), and reinforcement learning Wang et al. (2017). In this work, we focus on a more challenging version of the CO problem (1) that often arises in the DRO

formulation Haddadpour et al. (2022). Specifically, the problems that jointly minimize the summation of a compositional and a non-compositional objective. DRO has recently garnered significant attention from the research community because of its capability of handling noisy labels Chen et al. (2022), training fair machine learning models Qi et al. (2022), imbalanced Qi et al. (2020a) and adversarial data Chen & Paschalidis (2018). A standard approach to solve DRO is to utilize primal-dual algorithms Nemirovski et al. (2009) that are inherently slow because of a large number of stochastic constraints. The CO formulation enables the development of faster (dual-free) primal-only DRO algorithms Haddadpour et al. (2022). The majority of existing works to solve CO problems consider a centralized setting wherein all the data samples are available on a single server. However, modern large-scale machine-learning applications are characterized by the distributed collection of data by multiple clients Kairouz et al. (2021). This necessitates the development of distributed algorithms to solve the DRO problem.

Federated learning (FL) is a distributed learning paradigm that allows clients to solve a joint problem in collaboration with a server while keeping the data of each client private McMahan et al. (2017). The clients act as computing units where, within each communication round, the clients perform multiple updates while the server orchestrates the parameter sharing among clients. Numerous FL algorithms exist in the literature to tackle standard (non-compositional) optimization problems Li et al. (2019; 2020); Karimireddy et al. (2019); Sharma et al. (2019); Zhang et al. (2021); Khanduri et al. (2021); Karimireddy et al. (2020). However, there is a lack of efficient distributed implementations when it comes to CO problems. The major challenges in developing FL algorithms for solving the CO problem are:

[**C1**]. The compositional structure of the problem leads to *biased* stochastic gradient estimates, and this bias is amplified during local updates, which makes the analysis intractable Chen et al. (2021).

[**C2**]. Typically, data distribution at each client is different, resulting in *client drift* during local updates, which leads to divergence of federated CO algorithms. This is in sharp contrast to the standard FedAvg for non-CO problems, where client drift can be controlled Karimireddy et al. (2019).

[**C3**]. Majority of algorithms for solving CO rely on accuracy-dependent *large batch* gradients Huang et al. (2021); Haddadpour et al. (2022); Guo et al. (2022); Wu et al. (2024).

[**C4**]. The developed algorithms rely on the computation of expensive matrix (or vector) *projections* Gao (2024); Huang et al. (2023) or complex *multi-loop* structure Tarzanagh et al. (2022); Haddadpour et al. (2022); Huang et al. (2023).

These challenges naturally lead to the following question:

> *Can we develop FL algorithms that tackle* [**C1**] − [**C4**] *to solve CO in a federated setting?*

In this work, we address the above question and develop novel FL algorithms to solve CO problems. Although our development focuses on the DRO problem, the algorithms developed in our work have wider applicability to other general CO problems. The major contributions of our work are:

• We for the first time present a negative result that establishes that the vanilla FedAvg (customized to CO) is **incapable of solving** the CO problems as it leads to bias amplification during the local updates. This shows that either *additional communication/processing* or *additional assumptions* are required by FedAvg to mitigate the bias in the local gradient estimation.

• We develop two novel FL algorithms FedDRO and DS-FedDRO, for solving problems with both **compositional and non-compositional non-convex objectives**. To our knowledge, such algorithms have been absent from the open literature so far. Importantly, FedDRO and DS-FedDRO address the above-mentioned challenges by developing several key innovations in the algorithm design.

• FedDRO addresses [**C1**] − [**C4**] by adopting a **communication strategy** and **gradient estimators** that utilize the specific CO problem structure and allow us to control the gradient bias at the cost of additional (often low-dimensional) communication. In contrast, DS-FedDRO addresses these challenges by leveraging an additional structural assumption and introducing a **double-sided learning-rate** CO algorithm for FL, wherein the server-side aggregation mirrors the update dynamics of local clients.

• We establish the **convergence** of FedDRO and DS-FedDRO and show that to achieve an $\epsilon$-stationary point both algorithms require $\mathcal{O}(\epsilon^{-2})$ samples while achieving **linear speed-up** with the number of clients, i.e., requiring $\mathcal{O}(K^{-1}\epsilon^{-2})$ samples per client. Moreover, FedDRO achieves a communication complexity of $\mathcal{O}(\epsilon^{-3/2})$ when the inner compositional objective is low-dimensional and $\mathcal{O}(\epsilon^{-2})$ when it is high-dimensional, while DS-FedDRO achieves an improved communication complexity of $\mathcal{O}(\epsilon^{-1})$ irrespective of the dimensionality of the inner composite objective.

## 2 Problem setting

We focus on a general version of the CO problem defined in (1). We consider the following problem that often arises in DRO (see Section 2.1) in a FL setting with $K$ clients

$$\inf_{x \in \mathbb{R}^d} \left\{ \Phi(x) := h(x) + f(g(x)) \right\} \text{ with } h(x) := \frac{1}{K} \sum_{k=1}^{K} h_k(x) \text{ and } g(x) := \frac{1}{K} \sum_{k=1}^{K} g_k(x), \tag{2}$$

where each client $k \in [K]$ has access to the local functions $h_k : \mathbb{R}^d \to \mathbb{R}$ and $g_k : \mathbb{R}^d \to \mathbb{R}^{d_g}$ while $f(\cdot)$ is same as (1). The local functions $h_k(\cdot)$ and $g_k(\cdot)$ at each client $k \in [K]$ are: $h_k(x) = \mathbb{E}_{\xi_k \sim \mathcal{D}_{h_k}}[h_k(x; \xi_k)]$ and $g_k(x) = \mathbb{E}_{\zeta_k \sim \mathcal{D}_{g_k}}[g_k(x; \zeta_k)]$ and where $\xi_k \sim \mathcal{D}_{h_k}$ (resp. $\zeta_k \sim \mathcal{D}_{g_k}$) represents a sample of $h_k(\cdot)$ (resp. $g_k(\cdot)$) from distribution $\mathcal{D}_{h_k}$ (resp. $\mathcal{D}_{g_k}$).

In comparison to the basic CO (1), (2) is significantly challenging, first, because of the presence of both compositional and non-compositional objectives, and second, because of the distributed nature of the compositional function $g(\cdot)$. We would also like to point out that the algorithms and the analysis presented in this work can be easily extended to the problems where $f(\cdot) := 1/K \sum_{k=1}^{K} f_k(\cdot)$ is also distributed.

*Remark* 2.1 (Comparison to Gao et al. (2022) and Huang et al. (2021)). Formulation (2) is significantly different than the setting considered in Huang et al. (2021); Gao et al. (2022). Specifically, our formulation considers a setting where the compositional functions are distributed across agents, i.e., the function is $g = 1/K \sum_{k=1}^{K} g_k(x)$. In contrast, Huang et al. (2021); Gao et al. (2022) consider a setting with objective $1/K \sum_{k=1}^{K} f_k(g_k(\cdot))$, note here that the compositional function is local to each agent. This implies that algorithms developed in Huang et al. (2021); Gao et al. (2022) cannot solve problem (2). Importantly, problem (2) models a realistic FL setting while being more challenging compared to Huang et al. (2021); Gao et al. (2022) since in (2) the data heterogeneity of the inner problem also plays a role in the convergence of the FL algorithm. Please see the discussion in Appendix A.1 for more details.

### 2.1 Examples: CO reformulation of DRO problems

Here, we discuss different DRO formulations that can be efficiently solved using CO Haddadpour et al. (2022). DRO problem with $m$ training samples denoted as $\{\zeta_i\}_{i=1}^{m}$ is

$$\min_{x \in \mathbb{R}^d} \max_{\mathbf{p} \in P_m} \sum_{i=1}^{m} p_i \ell(x; \zeta_i) - \lambda D_*(\mathbf{p}, \mathbf{1}/m) \tag{3}$$

where $x \in \mathbb{R}^d$ is the model parameter, $P_m := \{\mathbf{p} \in R^m : \sum_{i=1}^{m} p_i = 1, p_i \geq 0\}$ is $m$-dimensional simplex, $D_*(\mathbf{p}, \mathbf{1}/m)$ is a divergence metric that measures distance between $\mathbf{p}$ and uniform probability $\mathbf{1}/m \in R^m$, and $\ell(x, \zeta_i)$ denotes the loss on sample $\zeta_i$, $\rho$ is a constraint parameter, and $\lambda$ is a hyperparameter. Next, we discuss two popular reformulations of (3) in the form of CO problems.

**DRO with KL-divergence.** Problem (3) is referred to as a KL-regularized DRO when the distance metric $D_*(\mathbf{p}, \mathbf{1}/m)$ is the KL-Divergence, i.e., we have $D_*(\mathbf{p}, \mathbf{1}/m) = D_{\mathrm{KL}}(\mathbf{p}, \mathbf{1}/m)$ with $D_{\mathrm{KL}}(\mathbf{p}, \mathbf{1}/m) := \sum_{i=1}^{m} p_i \log(p_i m)$. For this case, an equivalent reformulation of (3) is

$$\min_{x \in \mathbb{R}^d} \log \left( \frac{1}{m} \sum_{i=1}^{m} \exp \left( \frac{\ell(x; \zeta_i)}{\lambda} \right) \right), \tag{4}$$

which is a CO with $g(x) = 1/m \sum_{i=1}^{m} \exp(\ell(x; \zeta_i)/\lambda)$, $f(g(x)) = \log(g(x))$ and $h(x) = 0$.

**DRO with $\chi^2$- divergence.** Similar to KL-regularized DRO, (3) is referred to as a $\chi^2$-regularized DRO

Table 1: Comparison with the existing works. CO-ND refers to CO with a non-distributed compositional part (see Remark 2.1). CO + Non-CO refers to problems with both CO and Non-CO objectives. Multi-CO refers to a multi-level CO problem. VR refers to variance reduction. (I) and (O) refers to the inner and outer loop, respectively. †These focus on conditional CO problems (see Wu et al. (2024)), ‡Algorithm Fed-DR-SCGD relies on expensive matrix projections. *Theoretical guarantees for GCIVR exist only for the finite sample setting with $m$ total samples.

| ALGORITHM | SETTING | UPDATE | BATCH | COMP. | COMM. |
|---|---|---|---|---|---|
| ComFedL Huang et al. (2021) | CO-ND | SGD | $\mathcal{O}(\epsilon^{-2})$ | $\mathcal{O}(\epsilon^{-4})$ | $\mathcal{O}(\epsilon^{-2})$ |
| Local-SCGDM Gao et al. (2022) | CO-ND | Momentum SGD | $\mathcal{O}(1)$ | $\mathcal{O}(\epsilon^{-2})$ | $\mathcal{O}(\epsilon^{-1.5})$ |
| FCSG† Wu et al. (2024) | CO-ND | SGD | $\mathcal{O}(\epsilon^{-1})$ | $\mathcal{O}(K^{-1}\epsilon^{-3})$ | $\mathcal{O}(\epsilon^{-1.5})$ |
| Acc-FCSG-M† Wu et al. (2024) | CO-ND | VR | $\mathcal{O}(\epsilon^{-1})$ | $\mathcal{O}(K^{-1}\epsilon^{-2.5})$ | $\mathcal{O}(\epsilon^{-1})$ |
| FedNest Tarzanagh et al. (2022) | Bilevel | VR | $\mathcal{O}(1)$ | $\mathcal{O}(\epsilon^{-2})$ | $\mathcal{O}(\epsilon^{-2})$ |
| FedBiOAcc Li et al. (2024) | Bilevel | VR | $\mathcal{O}(1)$ | $\mathcal{O}(K^{-1}\epsilon^{-1.5})$ | $\mathcal{O}(\epsilon^{-1})$ |
| FedMBO Huang et al. (2023) | Bilevel | SGD | $\mathcal{O}(\ln(\epsilon^{-1}))$ | $\mathcal{O}(K^{-1}\epsilon^{-2})$ | $\mathcal{O}(\epsilon^{-2})$ |
| SimFBO Yang et al. (2024) | Bilevel | SGD | $\mathcal{O}(1)$ | $\mathcal{O}(K^{-1}\epsilon^{-2})$ | $\mathcal{O}(\epsilon^{-1})$ |
| Fed-DR-SCGD‡ Gao (2024) | Multi-CO | VR | $\mathcal{O}(1)$ | $\mathcal{O}(K^{-1}\epsilon^{-1.5})$ | $\mathcal{O}(\epsilon^{-1})$ |
| GCIVR* Haddadpour et al. (2022) | CO + Non-CO | VR | $\sqrt{m}$ (I), $m$ (O) | $\mathcal{O}(\sqrt{m}\epsilon^{-1} \wedge \epsilon^{-1.5})$ | $\mathcal{O}(\epsilon^{-1})$ |
| FedDRO (Ours) | CO + Non-CO | SGD | $\mathcal{O}(1)$ | $\mathcal{O}(K^{-1}\epsilon^{-2})$ | $\mathcal{O}(\epsilon^{-1.5})$ |
| DS-FedDRO (Ours) | CO + Non-CO | SGD | $\mathcal{O}(1)$ | $\mathcal{O}(K^{-1}\epsilon^{-2})$ | $\mathcal{O}(\epsilon^{-1})$ |

when $D_*(\mathbf{p}, \mathbf{1}/m)$ is the $\chi^2$-Divergence, i.e., we have $D_*(\mathbf{p}, \mathbf{1}/m) = D_{\chi^2}(\mathbf{p}, \mathbf{1}/m)$ with $D_{\chi^2}(\mathbf{p}, \mathbf{1}/m) := m/2 \sum_{i=1}^{m}(p_i - 1/m)^2$. For this case, an equivalent reformulation of (3) is

$$\min_{x\in\mathbb{R}^d} \frac{-1}{2\lambda m} \sum_{i=1}^{m} \left(\ell(x;\zeta_i)\right)^2 + \frac{1}{2\lambda}\left(\frac{1}{m}\sum_{i=1}^{m}\ell(x;\zeta_i)\right)^2 \tag{5}$$

with $g(x) = 1/m \sum_{i=1}^{m}\ell(x;\zeta_i)$, $f(g(x)) = g(x)^2/2\lambda$ and $h(x) = -\frac{1}{2\lambda m}\sum_{i=1}^{m}\left(\ell(x;\zeta_i)\right)^2$.

Both (4) and (5) can be restated in the FL setting of (2).

In addition to DRO, CO is used to model a variety of problems. Here, we highlight two representative examples: portfolio optimization Shapiro et al. (2021) and hyperparameter tuning Franceschi et al. (2018).

**Portfolio optimization.** In finance, portfolio optimization often involves balancing expected return and risk Shapiro et al. (2021). A classical formulation involves minimizing $\min_{x\in\mathcal{X}} -\mathbb{E}[R(x)] + \lambda\text{Var}(R(x))$. This formulation is widely used in mean-variance portfolio selection and other risk-aware financial optimization problems. Note that here the inner function $g(x) = R(x)$ denotes the return of a portfolio with weights $x$ and the outer function $f(\cdot) = -\mathbb{E}[\cdot] + \text{Var}(\cdot)$ captures statistical properties of the portfolio return while encoding the risk-return trade-off.

**Hyperparameter tuning.** Many meta-learning Finn et al. (2017) as well as hyperparameter optimization problems Franceschi et al. (2018) can be expressed in CO form as $\min_{\lambda\in\Lambda} L_{\text{val}}(g(\lambda))$, where the inner function $g(\lambda) = \arg\min_w L_{\text{train}}(w, \lambda)$ corresponds to the optimal model parameters obtained from a training problem, while the outer function $f(\cdot) = L_{\text{val}}(\cdot)$ evaluates the validation performance. Such formulations arise in hyperparameter tuning, neural architecture search, and dataset reweighting, where the validation loss depends implicitly on the solution of a lower-level training problem.

**Related work.** Please see Table 1 for a comparison of current approaches to solve CO problems in distributed settings. For a detailed review of centralized and federated non-convex CO and DRO problems, please see Appendix A. Here, we point out some drawbacks of the current approaches to solving federated CO problems:

• Do not guarantee linear speedup with the number of clients Huang et al. (2021); Haddadpour et al. (2022); Tarzanagh et al. (2022); Gao et al. (2022); Wu et al. (2024).

• Utilize complicated multi-loop algorithms with momentum or VR-based updates Tarzanagh et al. (2022) that sometime require computation of large batch size gradients Haddadpour et al. (2022) to guarantee convergence.

• Rely on expensive matrix (or vector) projections, restrictive assumptions, and sharing of additional sequences Gao (2024); Yang et al. (2024).

• Recently developed bilevel algorithms although in theory can be used to solve CO problems Tarzanagh et al. (2022); Li et al. (2024); Huang et al. (2023); Yang et al. (2024), however, since the algorithms are designed for bilevel problems they often have complicated structure, suffer from worse performance, and require sharing of additional parameters.

• Consider a restricted setting where the compositional objective is not distributed among nodes Huang et al. (2021); Gao et al. (2022); Wu et al. (2024).

Our work addresses all these issues and develops, FedDRO and DS-FedDRO, the first *simple* SGD-based FL algorithms to tackle CO problems with the distributed compositional objective.

## 3   Preliminaries

This section introduces the assumptions, definitions, and preliminary lemmas.

**Definition 3.1** (Lipschitzness). For all $x_1, x_2 \in \mathbb{R}^d$, a differentiable function $\Phi : \mathbb{R}^d \to \mathbb{R}$ is: Lipschitz smooth if $\|\nabla\Phi(x_1) - \nabla\Phi(x_2)\| \leq L_\Phi\|x_1 - x_2\|$ for some $L_\Phi > 0$; Lipschitz if $\|\Phi(x_1) - \Phi(x_2)\| \leq B_\Phi\|x_1 - x_2\|$ for some $B_\Phi > 0$ and; Mean-Squared Lipschitz if $\mathbb{E}_\xi\|\Phi(x_1; \xi) - \Phi(x_2; \xi)\|^2 \leq B_\Phi^2\|x_1 - x_2\|^2$ for some $B_\Phi > 0$.

We make the following assumptions on the local and global functions in the problem (2).

**Assumption 3.2** (Lipschitzness). The following holds
1. The functions $f(\cdot), h_k(\cdot), g_k(\cdot)$ for all $k \in [K]$ are differentiable and Lipschitz-smooth with constants $L_f, L_h, L_g > 0$, respectively.
2. The function $f(\cdot)$ and $h_k(\cdot)$ are Lipschitz with constants $B_f > 0$ and $B_h > 0$, respectively, and $g_k(\cdot)$ is mean-squared Lipschitz for all $k \in [K]$ with constant $B_g > 0$.

**Assumption 3.3** (Unbiased gradient and bounded variance). The stochastic gradients and function evaluations of the local functions satisfy

$$\mathbb{E}_{\xi_k}[\nabla h_k(x; \xi_k)] = \nabla h_k(x), \mathbb{E}_{\zeta_k}[\nabla g_k(x; \zeta_k)] = \nabla g_k(x),$$
$$\mathbb{E}_{\zeta_k}[g_k(x; \zeta_k)] = g_k(x), \mathbb{E}_{\zeta_k}[\nabla g_k(x; \zeta_k)\nabla f(y)] = \nabla g_k(x)\nabla f(y),$$
$$\mathbb{E}_{\xi_k}\|\nabla h_k(x; \xi_k) - \nabla h_k(x)\|^2 \leq \sigma_h^2, \mathbb{E}_{\zeta_k}\|\nabla g_k(x; \zeta_k) - \nabla g_k(x)\|^2 \leq \sigma_g^2, \mathbb{E}_{\zeta_k}\|g_k(x; \zeta_k) - g_k(x)\|^2 \leq \sigma_g^2,$$

for some $\sigma_h, \sigma_g > 0$ and for all $x \in R^d$ and $k \in [K]$.

**Assumption 3.4** (Bounded gradient heterogeneity). The heterogeneity of $h_k(\cdot)$ and $g_k(\cdot)$ is characterized as

$$\sup_{x \in \mathbb{R}^d} \|\nabla h_k(x) - \nabla h(x)\|^2 \leq \Delta_h^2 \text{ and } \sup_{x \in \mathbb{R}^d} \|\nabla g_k(x) - \nabla g(x)\|^2 \leq \Delta_g^2,$$

for some $\Delta_h, \Delta_g > 0$ for all $k \in [K]$.

The above assumptions are standard for non-convex CO problems. Specifically, Assumption 3.2 is required to establish Lipschitz smoothness of the $\Phi(\cdot)$ (see Lemma 3.5) and is standard in the analyses of CO problems Wang et al. (2017); Chen et al. (2021). Assumption 3.3 captures the effect of stochasticity in the gradient/function evaluations, while Assumption 3.4 characterizes the data heterogeneity among clients. This assumption is standard in FL, capturing non-IID client data while ruling out arbitrarily divergent local objectives Khanduri et al. (2021); Karimireddy et al. (2019). We would like to point out that in the assumption $\mathbb{E}_{\zeta_k}[\nabla g_k(x; \zeta_k)\nabla f(y)] = \nabla g_k(x)\nabla f(y)$ (see Assumption 3.3), $y$ may be correlated with the sample $\zeta_k$, and therefore, the unbiasedness assumption may not strictly hold. This assumption is made only for simplicity, similar to Chen et al. (2021), and can be easily avoided by drawing an independent sample instead of $\zeta_k$ so that the sample used is independent of $y$. Importantly, this modification does not affect the algorithm or the theoretical guarantees presented in this work. We note that all these assumptions are standard and have been utilized in the past to establish the convergence of many FL algorithms Yu et al. (2019a); Karimireddy et al. (2019); Zhang et al. (2021); Woodworth et al. (2020); Yang et al. (2022; 2021).

**Lemma 3.5** (Lipschitzness of $\Phi$). *Under Assumption 3.2 the compositional function, $\Phi(\cdot)$, defined in (2) is Lipschitz smooth with constant: $L_\Phi \coloneqq L_h + B_f L_g + B_g^2 L_f > 0$.*

Lemma 3.5 establishes Lipschitz smoothness of the compositional function $\Phi(\cdot)$. In general, $\Phi(\cdot)$ is a non-convex, and therefore, we cannot expect to globally solve (2). We instead rely on finding approximate stationary points of $\Phi(\cdot)$.

**Definition 3.6** ($\epsilon$-stationary point)**.** A point $x$ generated by an algorithm is an $\epsilon$-stationary point of $\Phi(\cdot)$ if $\mathbb{E}\|\nabla\Phi(x)\|^2 \le \epsilon$, where the expectation is taken w.r.t. the stochasticity of the algorithm.

**Definition 3.7** (Sample and communication complexity)**.** The sample complexity is the total (stochastic) gradient and function evaluations required to achieve an $\epsilon$-stationary solution. Similarly, communication complexity is the total number of communication rounds between the clients and the server required to achieve an $\epsilon$-stationary solution.

## 4 Federated non-convex CO algorithms

In this section, we for the first time establish the incapability of vanilla FedAvg to solve the distributed CO.

### 4.1 FedAvg fails to solve the federated CO problem

We show that vanilla FedAvg is not suitable for solving federated CO problems of form (2). To establish this, we consider a simple deterministic setting with $h(x) = 0$. For this setting, the local gradients of $\Phi(\cdot)$ are estimated using

$$\nabla\Phi_k(x) = \nabla g_k(x_k)\nabla f(y_k) \tag{6}$$

where sequence $y_k$ represents the local estimate of the inner function $g(x)$. To solve the above problem in a federated setup, we consider two candidate versions of FedAvg described in Case I and II of Algorithm 1. Similar to vanilla FedAvg, each agent performs multiple local updates within each communication round (Step 5 of Algorithm 1). Since $g(x) \coloneqq 1/k \sum_{k=1}^{k} g_k(x)$ with each agent $k \in [K]$ having access to only the local copy $g_k(\cdot)$, estimating $g(\cdot)$ locally within each communication round is not feasible. Therefore, each agent utilizes $y_k = g_k(x)$ as the local estimate of the inner function $g(\cdot)$. For communication, we consider two protocols. In the first setting, after $I$ local updates, in each communication round the agents share the locally updated parameters with the server and receive the aggregated parameter from the server (Case I in Step 7). In the second setting, in addition to the locally updated parameters the agents also share their local function evaluations $y_k^t = g_k(x_k^t)$ with the server and receive the aggregated embedding $\bar{y}^t$ from the server. This step is utilized to improve the local estimates of $g(\cdot)$ (Case II in Step 7). The algorithm executes for $\lfloor T/I \rfloor$ communication rounds.

---

**Algorithm 1** Vanilla FedAvg for non-convex CO

---

1: **Input**: Parameters: $\{\eta^t\}_{t=0}^{T-1}$, $I$

2: **Initialize**: $x_k^0 = \bar{x}^0$, $y_k^0 = \bar{y}^0$

3: **for** $t = 0$ to $T - 1$ **do**

4:     **for** $k = 1$ to $K$ **do**

5:        `Update:` $\begin{cases} \text{Compute } \nabla\Phi_k(x_k^t) \text{ using (6)} \\ \text{Local update: } x_k^{t+1} = x_k^t - \eta^t\nabla\Phi_k(x_k^t) \\ \text{Local composite function estimation: } y_k^{t+1} = g_k(x_k^{t+1}) \end{cases}$

6:        **if** $t + 1 \bmod I = 0$ **then**

7:            `[Case 1] Share:` $\begin{cases} \text{Global update: } x_k^{t+1} = \bar{x}^{t+1} \end{cases}$

              `[Case 2] Share:` $\begin{cases} \text{Global update: } x_k^{t+1} = \bar{x}^{t+1} \\ \text{Local composite function estimation: } y_k^{t+1} = g_k(\bar{x}^{t+1}) \\ \text{Global composite function estimation: } y_k^{t+1} = \bar{y}^{t+1} \end{cases}$

8:        **end if**

9:     **end for**

10: **end for**

---

In the following, we show that Algorithm 1 for both Cases I and II cannot solve the federated CO problem presented in (2) even in the simple deterministic setting with $h(x) = 0$.

**Theorem 4.1** (Vanilla FedAvg: Non-convergence for CO). *There exist functions $f(\cdot)$ and $g_k(\cdot)$ for $k \in [K]$ satisfying Assumptions 3.2, 3.3, and 3.4, and an initialization strategy such that for a fixed number of local updates $I > 1$, and for any $0 < \eta^t < C_\eta$ for $t \in \{0, 1, \dots, T-1\}$ where $C_\eta > 0$ is a constant, the iterates generated by Algorithm 1 under both Cases I and II do not converge to the stationary point of $\Phi(\cdot)$, where $\Phi(\cdot)$ is defined in (2) with $h(x) = 0$.*

Theorem 4.1 establishes that vanilla FedAvg is not suitable for solving federated CO problems. An important consequence of the above result is that under the set of Assumptions 3.2-3.4 the algorithms proposed in Huang et al. (2023); Gao (2024); Yang et al. (2024); Li et al. (2024) for solving federated CO (or bilevel) problems would fail to converge. We note that Theorem 4.1 highlights a fundamental mechanism that causes vanilla FedAvg to fail when applied to federated CO problems. The issue arises from the interaction of two effects: *(i)* the *inherent bias* in (stochastic) gradient estimators for compositional objectives because of the distributed nature of the objective, and *(ii) client drift* induced by performing multiple local updates. In the compositional setting, local updates propagate biased gradient estimates, and when the inner compositional objective is not synchronized at every communication round, this bias accumulates across rounds and cannot be controlled for any number of local steps $I > 1$. Specifically, Theorem 4.1 isolates this phenomenon and illustrates how such bias amplification prevents convergence. Moreover, regarding commonly used remedies, reducing the number of local steps or using techniques such as momentum, control variates, or variance reduction can mitigate client drift upto some extent, but do not eliminate the structural bias arising from the distributed inner compositional objective. As a result, the issue will still persist. This naturally leads to the question of how can we modify FedAvg such that it can efficiently solve CO problems of the form (2). Clearly, Theorem 4.1 suggests that sharing $y_k$'s in each iteration is required to ensure convergence of FedAvg since sharing the iterates $y_k$'s only intermittently leads to non-convergence of FedAvg. To this end, we propose to modify the FedAvg algorithm as presented in Algorithm 1 by sharing $y_k$ in each iteration $t \in \{0, 1, \dots, T-1\}$. The next result shows that the modified FedAvg resolves the non-convergence issue.

**Theorem 4.2** (Modified FedAvg: Convergence for CO). *Suppose we modify Algorithm 1 such that $y_k^t = \bar{y}^t$ is updated at each iteration $t \in \{0, 1, \dots, T-1\}$ instead of $[t+1 \mod I]$ iterations as in current version of Algorithm 1. Then if functions $f(\cdot)$ and $g_k(x)$ for $k \in [K]$ satisfy Assumptions 3.2, 3.3, and 3.4 such that for a fixed number of local updates $1 \leq I \leq \mathcal{O}(T^{1/4})$, there exists a choice of $\eta^t > 0$ for $t \in \{0, 1, \dots, T-1\}$ such that the iterates generated by (modified) Algorithm 1 converge to the stationary point of $\Phi(\cdot)$, where $\Phi(\cdot)$ is defined in (2) with $h(x) = 0$.*

### 4.2 A baseline federated non-convex CO algorithm: FedDRO

In this section, we propose a novel distributed non-convex CO algorithm, FedDRO, for solving (2). As demonstrated in Section 4.1, this problem is particularly challenging because of the compositional structure and heterogeneity of the local objectives. Motivated by Theorem 4.2 above, in this work we develop a novel approach where we utilize the structure of the CO problem to develop efficient FL algorithms for solving (2). Specifically, as also demonstrated in Section 2.1 we utilize the fact that the embedding $g(\cdot)$ is low-dimensional (e.g., $d_g = 1$), especially for DRO problems. This implies that sharing of $g(\cdot)$ will be relatively cheap in contrast to the high-dimensional model parameters of size $d$ which can be very large and take values in millions or even in billions for modern overparameterized neural networks Allen-Zhu et al. (2019). Therefore, like FedAvg, we share the model parameters intermittently after multiple local updates while sharing the low-dimensional embedding of $g(\cdot)$ frequently to handle the compositional objective. Please see Section 5 for an algorithm that avoids communicating this embedding, at the cost of an additional assumption and the use of a two-sided learning-rate scheme.

Moreover, to solve the CO problems for DRO the developed algorithms generally utilize batch sizes (for gradient/function evaluation) that are dependent on the solution accuracy Huang et al. (2021); Haddadpour et al. (2022). However, this is not feasible in most practical settings. In addition, to control the bias and to circumvent the need to compute large batch gradients, we utilize a momentum-based estimator to learn the inner function (see (8)) Chen et al. (2021). This construction allows us to develop FedAvg-type algorithms for solving non-convex CO problems wherein the local updates resemble the standard SGD updates.

The detailed steps of FedDRO are listed in Algorithm 2. During the local updates each client $k \in [K]$ updates its local model $x_k^t \ \forall t \in [T]$ using the local estimate of the stochastic gradients in Step 6. The stochastic gradient estimates for each client $k \in [K]$ are denoted by $\nabla\Phi_k(x_k^t; \xi_k)$ and are evaluated using the chain rule of differentiation as

$$\nabla\Phi_k(x_k^t; \xi_k^t) = \nabla h_k(x_k^t; \xi_k^t) + \nabla g_k(x_k^t; \zeta_k^t)\nabla f(\bar{y}^t) \tag{7}$$

where $\xi_k^t = \{\xi_k^t, \zeta_k^t\}$ represents the stochasticity of the gradient estimate. The variable $\bar{y}^t$ is designed to estimate the inner function $1/K \sum_{k=1}^{K} g_k(x)$ in (2). A standard approach to estimate $g_k(x)$ locally for each $k \in [K]$ is to utilize a large batch such that the gradient bias from the inner function estimate can be controlled Guo et al. (2022); Huang et al. (2021); Haddadpour et al. (2022). In contrast, we adopt a momentum-based estimate of $g_k(\cdot)$ at each client $k \in [K]$ that leads to a small bias asymptotically Chen et al. (2021). We note that the estimator utilizes a hybrid estimator that combines a SARAH Nguyen et al. (2017) and SGD Ghadimi & Lan (2013) estimate for the function values rather than the gradients Cutkosky & Orabona (2019). Specifically, individual $y_k^t$'s are estimated in Step 6 as

$$y_k^t = (1 - \beta^t)\left(y_k^{t-1} - g_k(x_k^{t-1}; \zeta_k^t)\right) + g_k(x_k^t; \zeta_k^t). \tag{8}$$

for all $k \in [K]$ and where $\beta^t \in (0, 1)$ is the momentum parameter. Motivated by the discussion in Section 4.1, the low-dimensional parameters $y_k^t \in \mathbb{R}^{d_g}$ are shared with the server after the $y_k^t$ update. The model parameters are then updated using the SG evaluated using (7). Finally, after $I$ local updates, the model parameters are aggregated on the server and shared with the clients after aggregation in Step 8. Next, we state the convergence guarantees.

**Convergence of FedDRO.** In the next theorem, we state the convergence of FedDRO.

**Theorem 4.3** (Convergence of FedDRO). *For Algorithm 2, choosing the step-size $\eta^t = \eta = \mathcal{O}(\sqrt{K/T})$, the momentum parameter $\beta = 4B_g^4 L_f^2 \eta$ for all $t \in \{0, 1, \ldots, T-1\}$, and $I = \mathcal{O}(T^{1/4}/K^{3/4})$. For $T \geq T_{th}$ where $T_{th}$ is defined in Appendix F, then under Assumptions 3.2, 3.3 and 3.4 for $\bar{x}^{a(T)}$ chosen According to Algorithm 2, we have*

$$\mathbb{E}\left\|\nabla\Phi(\bar{x}^{a(T)})\right\|^2 \leq \mathcal{O}\left(\frac{\mathcal{C}_{Sync}}{\sqrt{KT}}\right) + \mathcal{O}\left(\frac{\mathcal{C}_{Drift}}{\sqrt{KT}}\right),$$

*where constant $\mathcal{C}_{Sync}$ depends on initialization and $\mathcal{C}_{Drift}$ depends on the stochastic variance and data heterogeneity (See Appendix F).*

---

**Algorithm 2** FedDRO

---

1: **Input:** Parameters: $\{\beta^t\}_{t=0}^{T-1}$, $\{\eta^t\}_{t=0}^{T-1}$, $I$
2: **Initialize:** $x_k^{-1} = x_k^0 = \bar{x}^0$, $y_k^0 = \bar{y}^0$
3: **for** $t = 0$ to $T - 1$ **do**
4:     **for** $k = 1$ to $K$ **do**
5:         Local Update and Sharing:
6:         Update: $\begin{cases} \text{Compute } \nabla\Phi_k(x_k^t; \xi_k^t) \text{ using (7)} \\ \text{Local update: } x_k^{t+1} = x_k^t - \eta^t \nabla\Phi_k(x_k^t; \xi_k^t) \\ \text{Local composite function estimation: Compute } y_k^{t+1} \text{ using (8) ; share with server} \\ \text{Global composite function estimation: Receive } \bar{y}^{t+1} \text{ from server; update } y_k^{t+1} = \bar{y}^{t+1} \end{cases}$
7:         **if** $t + 1 \bmod I = 0$ **then**
8:             Aggregate : $\begin{cases} \text{Global update: } x_k^{t+1} = \bar{x}^{t+1} \end{cases}$
9:         **end if**
10:     **end for**
11: **end for**
12: **Return:** $\bar{x}^{a(T)}$ where $a(T) \sim \mathcal{U}\{1, \ldots, T\}$.

---

We note that the condition on $T \geq T_{\text{th}}$ is required only for theoretical purposes. Specifically, it ensures that the step-size $\eta = \mathcal{O}(\sqrt{K/T})$ is upper-bounded. A similar requirement has also been posed in Yu et al. (2019a;b); Khanduri et al. (2021) in the past. Theorem 4.3 captures the effect of heterogeneity, stochastic variance, and the initialization on the performance of FedDRO. Theorem 4.3 also states that there exists a choice of the number of local updates that guarantee that FedDRO achieves the same convergence performance as a standard FedAvg Karimireddy et al. (2019); Woodworth et al. (2020); Yu et al. (2019a); Khanduri et al. (2021) for solving the non-CO problems. Next, we characterize the sample and communication complexities of FedDRO.

**Corollary 4.4** (Computation and communication)**.** *Under the setting of Theorem 4.3 the following holds for FedDRO*

*(i) The **sample complexity** is $\mathcal{O}(\epsilon^{-2})$. This implies that each client requires $\mathcal{O}(K^{-1}\epsilon^{-2})$ samples to reach an $\epsilon$-stationary point achieving linear speed-up.*

*(ii) The **communication complexity** of is $O(\epsilon^{-3/2})$.*

*Remark* 4.5 (Performance with dimension of $g(\cdot)$). The sample and communication complexities guaranteed by Corollary 4.4 match those of the standard FedAvg algorithm Yu et al. (2019b) for stochastic non-convex non-CO problems and improve upon the guarantees of existing federated CO algorithms Huang et al. (2021); Gao et al. (2022); Guo et al. (2022). In addition to the $\mathcal{O}(\epsilon^{-3/2})$ communication complexity associated with sharing high-dimensional model parameters, FedDRO also communicates $\mathcal{O}(\epsilon^{-2}K^{-1}d_g)$ typically low-dimensional embeddings (often scalar quantities, as illustrated in Section 2.1), where $d_g$ denotes the dimension of $g(\cdot)$. Consequently, the total number of real-valued quantities transmitted by each client during the execution of FedDRO is $\mathcal{O}(\epsilon^{-3/2}d + \epsilon^{-2}K^{-1}d_g)$, where $d$ denotes the model dimension. Observe that if $d \geq \epsilon^{-1/2}K^{-1}d_g$, the overall complexity is still dominated by the first term, yielding the same $\mathcal{O}(\epsilon^{-3/2})$ rate. However, if $d_g \geq \sqrt{\epsilon}Kd$, the second term becomes dominant, and the communication complexity degrades to $\mathcal{O}(\epsilon^{-2})$, which matches the classical distributed SGD guarantee. Nevertheless, we note that for many problems of practical interest, including DRO Haddadpour et al. (2022), portfolio optimization Shapiro et al. (2021), and phase retrieval Duchi & Ruan (2019), the function $g(\cdot)$ is typically low-dimensional.

However, to make FedDRO fully federated it is desirable to develop an algorithm that can circumvent the need to communicate the sequences $y_k^t$ at each time instant. Next, we tackle this challenge and develop a novel two-sided learning rate algorithm DS-FedDRO.

---

**Algorithm 3** DS-FedDRO

1: **Input**: Parameters: $\{\beta^t\}_{t=0}^{T-1}$, $\{\eta^t\}_{t=0}^{T-1}$, $I$, $\gamma_x$, $\gamma_y$
2: **Initialize**: $x_k^{-1} = x_k^0 = x^\tau$, $y_k^0 = y^\tau$ with $\tau = 0$, $\forall k \in [K]$
3: **for** $t = 0$ to $T - 1$ **do**
4:     **for** $k = 1$ to $K$ **do**
5:         Update:$\begin{cases} \text{Compute } \nabla\Phi_k(x_k^t; \xi_k^t) \text{ using (7)} \\ \text{Local update: } x_k^{t+1} = x_k^t - \eta^t \nabla\Phi_k(x_k^t; \xi_k^t) \\ \text{Local composite function estimation: } y_k^{t+1} = (1-\beta^t)y_k^t + \beta^t g_k(x_k^{t+1}; \zeta_k^{t+1}) \end{cases}$
6:         **if** $t + 1 \mod I = 0$ **then**
7:             Share:$\begin{cases} \text{Global update: } x^{\tau+1} = x^\tau - \frac{\gamma_x^\tau}{K}\sum_{k=1}^K (x^\tau - x_k^{t+1}) \\ x_k^{t+1} = x^{\tau+1}, \quad \forall k \in [K] \\ \text{Global composite function estimation: } y^{\tau+1} = y^\tau - \frac{\gamma_y^\tau}{K}\sum_{k=1}^K (y^\tau - y_k^{t+1}) \\ y_k^{t+1} = y^{\tau+1}, \quad \forall k \in [K] \end{cases}$
8:             $\tau = \tau + 1$
9:         **end if**
10:     **end for**
11: **end for**
12: **Return:** $\bar{x}^{a(T)}$ where $a(T) \sim \mathcal{U}\{1, ..., T\}$.

---

# 5 Federated non-convex CO with two-sided learning rates: DS-FedDRO

In this section, we propose a novel algorithm called DS-FedDRO (FedDRO with double-sided learning rates) that relies on the two-sided learning rate utilized in classical FL algorithms to improve both the experimental and the theoretical performance Yang et al. (2021); Reddi et al. (2020). Importantly, we show that, under the additional Assumption 5.1, DS-FedDRO completely avoids the communication of sequence $y_k^{t+1}$ as required by FedDRO while at the same time achieving improved communication complexity. The steps of DS-FedDRO are listed in Algorithm 3. Let us point out a few key differences compared to FedDRO. First, note in Step 7 that instead of performing simple aggregation, the algorithm relies on a two-sided learning rate update rule for both the $x$- and the $y$-update. Second, note that the two-sided learning update rule also allows us to update the sequence $y$ utilizing only a single stochastic gradient computation in Step 5. In contrast, FedDRO required two stochastic gradient computations to update $y$. In effect, DS-FedDRO, not only reduces the communication complexity but also improves the per iteration computation complexity over FedDRO. In the following, we present the convergence guarantees of DS-FedDRO and contrast them to those achieved by FedDRO.

## 5.1 Main results: Convergence of DS-FedDRO

For the theoretical results of this section, we utilize a different notion of heterogeneity compared to Assumption 3.4 presented earlier.

**Assumption 5.1** (Bounded functional heterogeneity)**.** The heterogeneity of $g_k(\cdot)$ is characterized as $\sup_{x \in R^d} \|g_k(x) - g(x)\|^2 \leq \Delta_g^2$, for some $\Delta_g > 0$ and for all $k \in [K]$.

Assumption 5.1 is a weaker version of (Gao, 2024, Assumption 3.3), where bounded heterogeneity is imposed directly on the stochastic objectives. It is also closely related to assumptions used in bilevel optimization with quadratic lower-level objectives (Huang et al., 2023; Yang et al., 2024, Assumption 5). Although restrictive, this type of condition is common in the optimization literature and is motivated by the bounded gradient heterogeneity assumptions widely used in federated learning (Yu et al., 2019b; Karimireddy et al., 2019; Zhang et al., 2021). We now state the main result.

**Theorem 5.2.** *For Algorithm 3, choosing the local step-sizes $\eta^t = \eta = \mathcal{O}(\sqrt{1/TI})$ and the momentum parameter $\beta^t = \beta = c_\beta \eta$ for all $t \in \{0, 1, \ldots, T-1\}$. Choosing the server step-sizes $\gamma_x = \mathcal{O}(\sqrt{K/T})$, $\gamma_y = c_{\gamma_y}\gamma_x$, and $I = \mathcal{O}(\sqrt{T/K})$. Then under Assumptions 3.2, 3.3, 3.4, and 5.1 for $\bar{x}^{a(T)}$ chosen according to Algorithm 3, we have*

$$\mathbb{E}\big\|\nabla\Phi(\bar{x}^{a(T)})\big\|^2 \leq \mathcal{O}\bigg(\sqrt{\frac{\mathcal{C}_{Sync}}{KT}}\bigg) + \mathcal{O}\bigg(\frac{\mathcal{C}_{Drift}}{T}\bigg)$$

*for some constants $c_\beta$, $c_{\gamma_y}$, $\mathcal{C}_{Sync}$ and $\mathcal{C}_{Drift}$.*

Next, we characterize the sample and communication complexity of DS-FedDRO.

**Corollary 5.3** (Computation and communication)**.** *Under the setting of Theorem 5.2 the following holds for DS-FedDRO*

*(i) The **sample complexity** is $\mathcal{O}(\epsilon^{-2})$. This implies that each client requires $\mathcal{O}(K^{-1}\epsilon^{-2})$ samples to reach an $\epsilon$-stationary point achieving linear speed-up.*

*(ii) The **communication complexity** is $O(\epsilon^{-1})$.*

First, note that DS-FedDRO in addition to achieving linear speed-up also improves the communication performance compared to FedDRO. Moreover, it is important to note that the communication complexity of $O(\epsilon^{-1})$ matches the best-known communication complexity even for standard FL problems Zhang et al. (2021); Acar et al. (2020). Moreover, compared to federated CO algorithms Gao (2024); Haddadpour et al. (2022) and bilevel optimization algorithms Tarzanagh et al. (2022); Yang et al. (2024); Li et al. (2024); Huang et al. (2023) the update rules employed by DS-FedDRO (and FedDRO) are much simpler, computation efficient (does not require any projection) and require the sharing of fewer sequences, thereby, making DS-FedDRO efficient compared to such algorithms.

*Remark* 5.4 (Comparison of DS-FedDRO to FedDRO). Although DS-FedDRO performs significantly better compared to FedDRO in terms of communication performance, there are some drawbacks of DS-FedDRO: *(i) Additional tuning parameters.* From a practical perspective, because of the addition of server-side learning rates for both $x$- and $y$- updates, DS-FedDRO requires more parameters to tune compared to FedDRO. *(ii) Additional assumption.* From a theoretical perspective, the improved performance of DS-FedDRO is made possible with additional Assumption 5.1 compared to FedDRO which did not require Assumption 5.1.

## 6 Experiments

In this section, we evaluate the performance of FedDRO and DS-FedDRO with both centralized and distributed baselines. Our goal is to *1)* establish the superior performance of FedDRO and DS-FedDRO compared to popular federated DRO baselines, and *2)* evaluate the performance of FedDRO and DS-FedDRO with different numbers of local updates to capture the effect of data heterogeneity. To evaluate the performance of FedDRO and DS-FedDRO, we focus on two tasks: classification with an imbalanced dataset and learning with fairness constraints. For the first task, we use CIFAR10-ST and CIFAIR-100-ST datasets Qi et al. (2020b) (unbalanced versions of CIFAR10 and CIFAR100 Krizhevsky et al. (2009)) for image classification, and the performance is measured by training and testing accuracy achieved by different algorithms. For the second task, we use the Adult dataset Dua & Graff (2017) for enforcing equality of opportunity (on protected

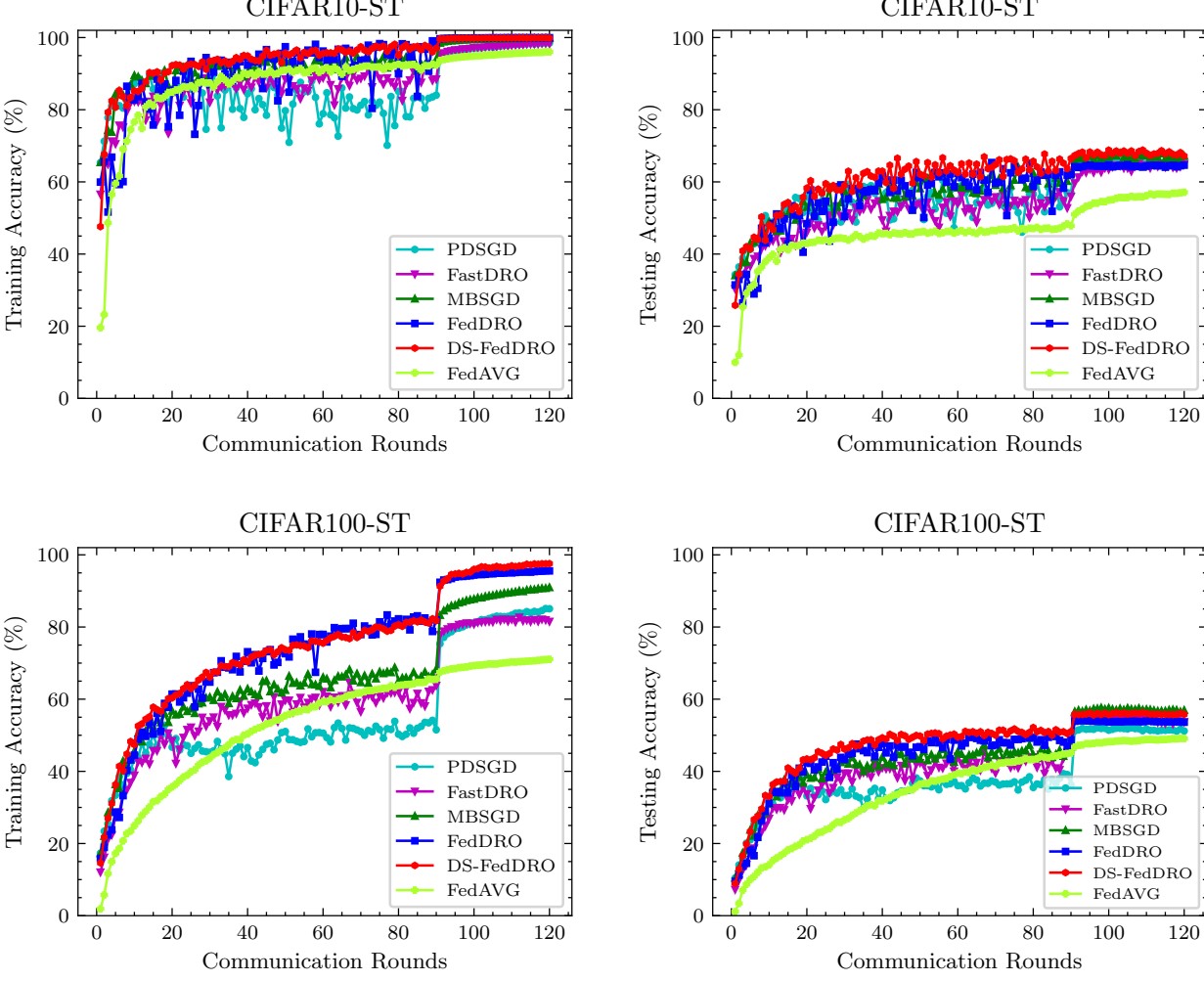

Figure 1: Performance comparison of various algorithms on CIFAR10-ST and CIFAR100-ST datasets.

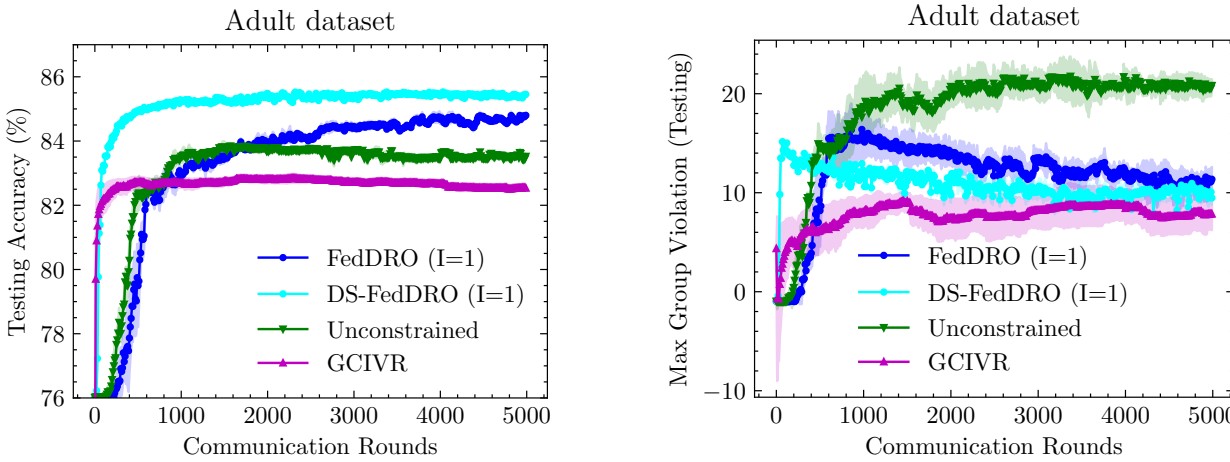

Figure 2: Comparison of FedDRO, DS-FedDRO, GCIVR, and the unconstrained baseline on Adult dataset.

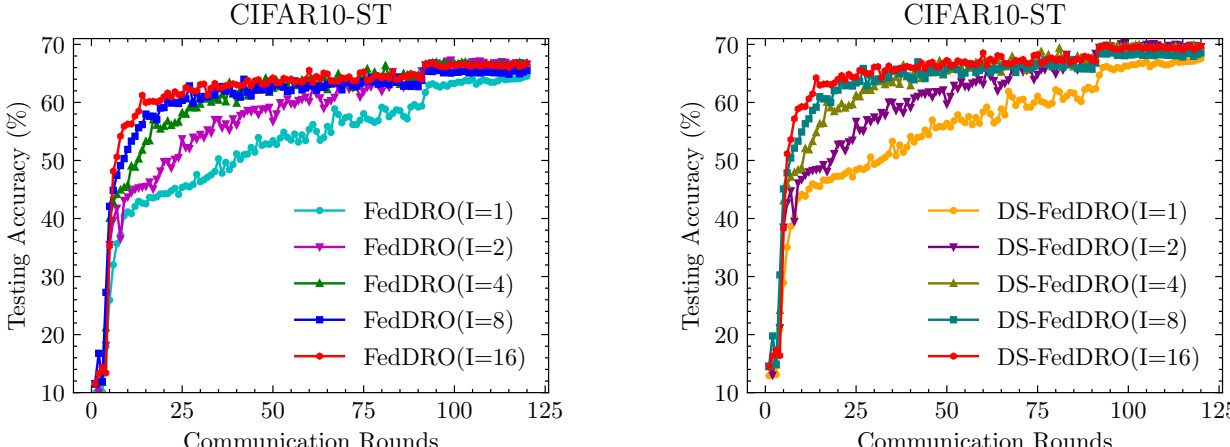

Figure 3: FedDRO and DS-FedDRO on the CIFAR10-ST and CIFAR100-ST for different $I$.

classes) on tabular data classification Hardt et al. (2016). For this setting, the performance is evaluated by training/testing accuracy, and the constraint violations, which are measured by the gap between the true positive rate of the overall data and the protected groups Haddadpour et al. (2022). Please see Appendix B for further details and additional experiments.

**Baseline methods.** For the CIFAR10-ST and CIFAR100-ST datasets we compare FedDRO and DS-FedDRO with popular centralized and distributed baselines for classification with imbalanced data. The baselines adopted for comparison are a popular DRO method, FastDRO Levy et al. (2020), a primal-dual SGD approach to solve constrained problems with many constraints, PDSGD Xu (2020), classical FedAvg (see Algorithm 2 Case II) Yu et al. (2019b;a) and a popular baseline minibatch SGD, MBSGD, customized for CO Ghadimi & Lan (2013). For the adult dataset, we use GCIVR Haddadpour et al. (2022) as the baseline distributed model to compare with FedDRO and DS-FedDRO, since like these, it is the only algorithm that can deal with CO and non-CO objectives simultaneously. We also implement parallel SGD (serving as a proxy for FedAvg with a single local update). as a baseline that ignores the fairness constraints, referred to as unconstrained in the experiments.

**Implementation details.** We use 8 clients to model the distributed setting and split the (unbalanced) dataset equally for each client (for performance with 32 clients please refer to the Appendix). We use ResNet20 for classification tasks on CIFAR10-ST and CIFAR100-ST datasets. For a fair comparison with centralized baselines, we choose $I = 1$ for FedDRO and implement a parallel version of the centralized algorithms where the overall gradient computation is $K$ times larger for each algorithm. This is to make sure that the overall

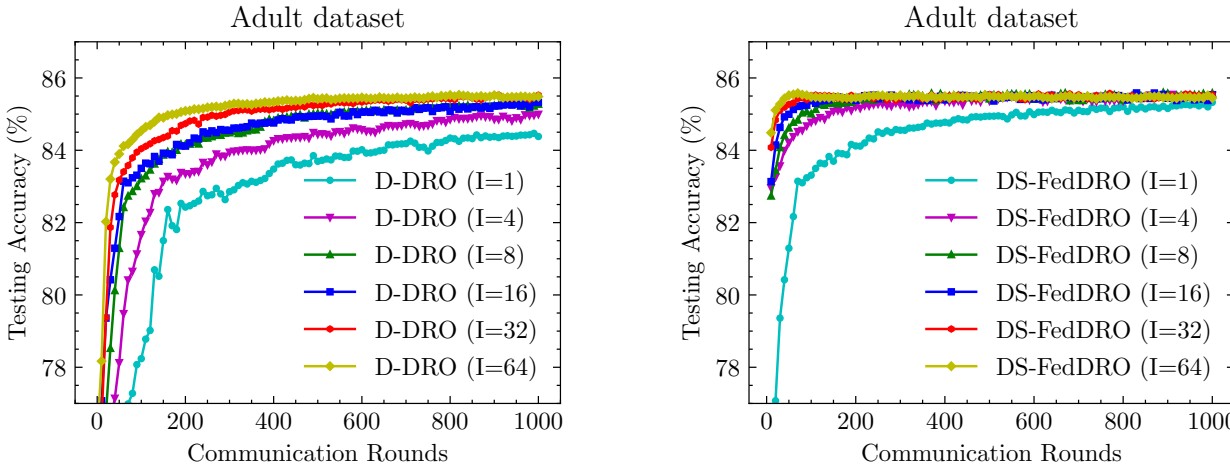

Figure 4: Performance of FedDRO and DS-FedDRO on Adult dataset for different $I$.

gradient computations in each step are uniform across all algorithms. Performance with different values of $I$ is evaluated separately. For each algorithm, we used a batch size of 16 per client, and the learning rates were tuned from the set $\{0.001, 0.01, 0.05, 0.1\}$, the learning rate was dropped to $1/10^{\text{th}}$ after 90 communication rounds. As for the two-sided learning rates for DS-FedDRO we select 1.3 and 1.4 for the respective tasks. For fairness-constrained classification on the Adult dataset, we use a logistic regression model. For this experiment, we adopt the settings suggested in Haddadpour et al. (2022), for FedDRO and DS-FedDRO we keep the same setting as in the earlier task. Each experiment was repeated three times using different random seeds, and we report the mean and standard deviation of the results.

**Discussion.** In Figure 1, we evaluate the performance of FedDRO and DS-FedDRO against parallel implementations of centralized and distributed baselines on unbalanced CIFAR datasets. Both methods achieve higher training accuracy and comparable or better test accuracy relative to the baselines. Notably, DS-FedDRO slightly outperforms FedDRO, which highlights the benefit of employing two-sided learning rates at both the client and server levels. This design allows the server to more effectively aggregate and correct the updates received from heterogeneous clients, leading to improved optimization stability. In contrast, FedAvg performs significantly worse on these compositional objectives. This observation is consistent with our theoretical analysis of Section 4.1 showing that vanilla FedAvg fails to converge for federated CO problems because it does not properly account for the nested gradient structure. In Figure 2, we compare FedDRO and DS-FedDRO with the distributed baseline GCIVR on the Adult dataset. Both methods achieve higher test accuracy while maintaining comparable constraint violation performance as the number of communication rounds increases. These results demonstrate that our methods can effectively balance optimization accuracy and constraint satisfaction while maintaining communication efficiency.

Finally, in Figures 3 and 4, we examine the effect of the number of local updates, $I$, on the CIFAR10 and Adult datasets, respectively. As $I$ increases, both FedDRO and DS-FedDRO initially exhibit improved performance due to reduced communication overhead and more effective utilization of local computation. However, beyond a certain point, the performance saturates. This phenomenon arises from client drift caused by data heterogeneity across clients: when too many local updates are performed, the local models deviate from the global objective, which limits the benefits of further local computation. In particular, moderate values of $I$ provide the best trade-off between communication efficiency and optimization stability, whereas excessively large values of $I$ lead to diminishing returns due to increased client drift. This empirical observation is consistent with our theoretical analysis (cf. Theorems 4.3 and 5.2), which shows that increasing the number of local updates accelerates optimization only up to a certain threshold, beyond which the effect of heterogeneity dominates and the improvement saturates.

## 7    Conclusion

This work establishes, for the first time, that vanilla FedAvg fails to solve CO problems in federated learning under standard assumptions. To overcome this limitation, we show that convergence of SGD-based FL algorithms for CO problems of the form (2) requires sharing additional, typically low-dimensional embeddings of the stochastic compositional objective. Building on this insight, we propose FedDRO, a federated CO algorithm that achieves linear speedup in the number of clients without large batch sizes. We further develop DS-FedDRO, which, under an additional assumption and a double-sided learning rate, eliminates the need for extra communication while attaining improved communication complexity.

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
