# OpenReview forum: "On Federated Compositional Optimization: Algorithms, Analysis, and Guarantees"
_TMLR — Accepted by TMLR_

### Review · Reviewer_K18Q · 2026-02-02

**Summary Of Contributions:**

The paper proposes two Fed-Avg type algorithms (FedDRO and DS-FedDRO) for non convex Compositional optimization in FL settings. The authors consider the Distributionally Robust Optimization (DRO), focusing on a more challenging version: a problem that minimizes the summation of compositional and non compositional objective. This problem is considered in an FL scenario. Current CO solutions cannot be directly used in FL.
First, the authors theoretically show that FedAvg fails to converge and cannot solve the federated CO problem.
To solve this, they introduce a first algorithm called FedDRO that adopts a momentum-based estimate of $g_k$ at each client. FedDRO achieves a sample complexity of $O(\epsilon^{-2})$.
The second algorithm is DS-FedDRO, which uses a two-sided learning rate, reducing the communication complexity and improving the per-iteration computation complexity $O(\epsilon^{-1})$ compared with $O(\epsilon^{-\frac{3}{2}})$ of FedDRO. At the same time, DS-FedDRO requires more hyperparameters to tune compared with FedDRO.
For both methods, the author provides a convergence analysis and comparison with other methods.

**Audience:**

Yes

**Audience Explanation:**

The paper is about Federated Learning and Compositional Optimization, which are interesting areas for the audience of TMLR.

**Broader Impact Concerns:**

I do not have any ethical concerns about this work.

**Claims And Evidence:**

No

**Claims Explanation:**

The theoretical part of the paper convinced me about the proposed approach. Both the Vanilla FedAvg non-convergence (Theorem 4.1) and the convergence of FedDRO and DS-FedDRO (Theorems 4.3 and 5.1) are supported by proofs in the Appendix.
However, the experimental part of the paper has several problems. The authors claim that their goal in the experiments is to establish the superior performance of their algorithms and evaluate the performance of FedDRO with different numbers of local updates.

1) In Figure 1, the authors used different y-axis limits: the first plot (training accuracy with Cifar10-ST) has a y-axis between 40% and 100%, while the second plot, which shows the testing accuracy, has a y-axis limit between 20% and 70%. Something similar happens for the third and fourth plots. This not only shows a comparison with different scales but also shows a drop in the accuracy between train and test. Could you please elaborate on this?

2) From the main paper, it is not clear what FL setup was used in the experiments. How is the data distributed across the clients? How are the hyperparameters tuned?

3) In the experiments, the authors used 8 clients for the FL simulation. In general, it is common to show experiments with more than 8 clients to show how the methods scale with increasing clients. Would it be possible to include experiments with more clients?

4) There is no discussion of the results; the authors are basically just explaining what is inside the plots without stating any conclusion or "take-home message."

5) All the plots are hard to read. The legend, x-axis, and y-axis are too small. The different lines overlap, and it's hard to understand them. In some of them, there is a shadow around the lines. Please fix plot scales and readability.

6) Since DS-FedDRO should have better communication complexity than FedDRO, would it be possible to have an empirical test of this with a dedicated plot to compare the communication needed for the two methods?

7) In Figure 3,  we observe that as I increases, the performance of the two algorithms improves. Specifically, line with I=16 performs better than I=1. In Section 1, points [C1] and [C2], the authors wrote that the compositional structure leads to biased gradient estimates and that "this bias is amplified during local updates". If the local updates cause failures due to bias amplification, I would expect that the setting with I=1 would be the most accurate. Could authors please better explain the results of Figure 3 and elaborate more on this?

8) Is the plots, is the line, the average of some different runs, and the shadow the std? Could you specify this in the paper, please?

9) In Figure 1, I was also expecting to see a comparison with the standard FedAvg rather than a comparison only with centralised baselines. Would it be possible to add it?

**Requested Changes:**

I would appreciate the authors addressing my concerns written above.

---

### Review · Reviewer_HFCi · 2026-02-10

**Summary Of Contributions:**

This paper tackles federated nonconvex compositional optimization where the inner function is distributed across clients, a more general setting than prior federated CO work. It first proves a negative result that vanilla FedAvg-style adaptations can fail to converge due to bias amplification under heterogeneity, then proposes two simple FedAvg-type algorithms: FedDRO, which restores convergence by frequently synchronizing a low-dimensional inner-function estimate, and DS-FedDRO, which further improves communication via a two-sided learning-rate scheme under an additional heterogeneity assumption. The authors provide convergence guarantees and report empirical gains on DRO-style tasks including imbalanced classification and fairness constraints.

**Audience:**

Yes

**Audience Explanation:**

Some researchers in federated learning or optimizaitoin will get interested into this paper.

**Claims And Evidence:**

No

**Claims Explanation:**

**1) Strong / potentially unrealistic assumptions (Assumption 3.2)**

The convergence analysis relies on a bundle of global regularity conditions (smoothness + Lipschitz + mean-squared Lipschitz) in Assumption 3.2. The paper asserts these are “standard,” but does not demonstrate that they hold for the motivating federated learning settings (e.g., deep networks in the experiments).

**2) Questionable unbiasedness requirement in Assumption 3.3 (correlation issue)**

Assumption 3.3 requires $\mathbb{E}_{\zeta_k}[\nabla g_k(x;\zeta_k)\nabla f(y)] = \nabla g_k(x)\nabla f(y)$. In the proposed algorithms, $y$ is constructed using past stochastic samples and is generally **correlated** with $\zeta_k$, so this identity is unclear and may fail in realistic scenarios.

**3) Limited insight and generality of the “FedAvg fails” message**

Although Theorem 4.1 establishes non-convergence of vanilla FedAvg in a constructed setting, the manuscript does not clearly articulate a general mechanism beyond “bias amplification,” nor does it discuss whether the failure persists under common remedies (e.g., decreasing local steps, control variates, adaptive synchronization).

**4) Narrow applicability due to reliance on “low-dimensional \(g\)” communication**

A central design choice in FedDRO is frequent communication of the embedding $y_t \approx g(x_t)$, justified by assuming $g$ is low-dimensional (often scalar in DRO). This significantly limits applicability: in many compositional objectives (and in deep learning variants), $\dim(g)$ may be large, making the proposed communication scheme expensive or impractical.

**5) DS-FedDRO relies on an additional strong heterogeneity assumption (Assumption 5.1)**

DS-FedDRO’s improved communication complexity depends on Assumption 5.1, a uniform bound on $\sup_x \|g_k(x)-g(x)\|^2$, which can be quite restrictive in heterogeneous FL and is not validated empirically. The paper provides no evidence that this assumption can be satisfied with interesting examples or that the method is robust when it fails.

**Requested Changes:**

Considering the five concerns I raised above, I suggest the authors provide further clarification below:

1. Please provide at least one concrete end-to-end application example (with explicit \(f,g_k,h_k\)) where **all** assumptions can be verified, or relax/replace the assumptions with conditions that better match modern FL practice.

2. Clarify precisely under what conditions the equality $\mathbb{E}_{\zeta_k}[\nabla g_k(x;\zeta_k)\nabla f(y)] = \nabla g_k(x)\nabla f(y)$ holds (e.g., independence/conditioning structure), provide nontrivial examples, or modify the analysis to avoid requiring such a condition.

3. Strengthen the conceptual explanation of failure (what exactly blows up and why), and discuss whether there are variants of FedAvg can work or not

4. Provide either (i) a clear characterization of how costs/guarantees degrade with $\dim(g)$, or (ii) additional experiments/settings beyond scalar-DRO where $\dim(g)$ is nontrivial.

5. Discuss realistic scenarios where Assumption 5.1 is expected to hold; alternatively, provide robustness analysis when this assumption is violated.

I believe the paper could be suitable for acceptance provided the above requests are adequately addressed.

---

### Review · Reviewer_yoMz · 2026-02-28

**Summary Of Contributions:**

This paper studies a compositional optimization problem in a federated learning setting, where multiple clients have heterogeneous data and the objective depends on the average of their local losses.

The authors first show that the standard FedAvg algorithm does not work for this problem (it diverges) without modifications. Based on this, they design a new algorithm that uses more frequent communication between the clients and the server for some parameters that are usually low dimensional. They also use momentum-based variance reduction (MVR) to avoid using very large batch sizes.

They further propose an extension of the algorithm that avoids the extra communication under an additional assumption, using a two-sided stepsize.

**Audience:**

Yes

**Audience Explanation:**

This paper would be of interest to researchers working on federated learning, as it studies a non-trivial problem that cannot be easily solved by existing FL algorithms and proposes a modified method to address it. It may also interest researchers working on compositional optimization, as it shows how such problems can be handled in a federated learning setting and could be extended or improved in future work.

**Claims And Evidence:**

Yes

**Claims Explanation:**

The statements of the theorems and lemmas appear clear and reasonable. I briefly checked the proofs, and they seem correct, although I did not verify them in full detail.

**Requested Changes:**

- Could you provide more examples and motivation for the use cases of the CO problem?
- Could you elaborate more on existing federated CO algorithms and explain how they differ from your method and in what sense they perform worse?

---

### Decision · Action_Editor_3v1w · 2026-04-18

**Recommendation:** Accept with minor revision

**Additional Comments:**

The revised version has improved meaningfully, and two reviewers ended with leaning-accept recommendations. The main remaining concern from the other reviewer is the experimental section and the visualization of the results, which were still considered not strong enough for journal publication. I agree that the figures should be improved further in the final version, ideally using vector graphics (e.g., PDF) rather than pixel-based formats, with font sizes and overall formatting consistent with the main text. Another concern raised in the reviews is that the assumptions, although clearly stated, are not yet fully justified in terms of practical plausibility.

**Audience:**

Yes

**Audience Explanation:**

Yes. The paper studies a technically meaningful problem at the intersection of federated learning and compositional optimization. The proposed algorithms and guarantees could be of interest to researchers working on federated optimization, compositional optimization, and related distributed learning problems.

**Claims And Evidence:**

Yes

**Claims Explanation:**

This paper studies a meaningful federated compositional optimization problem, and its assumptions are clearly stated. The reviewers found the main claims credible. The remaining concerns focus mainly on the practical plausibility of some assumptions and on the strength and presentation of the experimental section, rather than on a specific identified flaw in the main analysis.